# Seroprevalence of Immunoglobulin G Antibodies Against *Mycobacterium avium* subsp. *paratuberculosis* in Dogs Bred in Japan

**DOI:** 10.3390/vetsci7030093

**Published:** 2020-07-17

**Authors:** Takashi Kuribayashi, Davide Cossu, Eiichi Momotani

**Affiliations:** 1Laboratory of Immunology, School Life and Environmental Science, Azabu University, Kanagawa 252-5201, Japan; 2Department of Neurology, School of Medicine, Juntendo University, Tokyo 113-8431, Japan; davide@juntendo.ac.jp; 3Comparative Medical Research Institute, Ibaraki 305-0856, Japan; eiichimomotani@gmail.com

**Keywords:** MAP, dogs, IgG antibody, Johne’s disease

## Abstract

In this study, the seroprevalence of immunoglobulin G (IgG) antibodies against *Mycobacterium avium* subsp. *paratuberculosis* (MAP) in dogs bred in Japan was evaluated. Ninety-two non-clinical samples were obtained from three institutes and fifty-seven clinical samples were obtained from a veterinary hospital in Japan. Serum titers of total IgG, IgG_1_ and IgG_2_ isotype antibodies against MAP were measured using an indirect enzyme-linked immunosorbent assay (ELISA). The IgG antibodies against MAP in non-clinical serum obtained from three institutes was observed to be 2.4%, 20% and 9.0%. Similarly, the IgG_1_ antibodies titers against MAP were observed to be 7%, 20% and 0%. Lastly, the IgG_2_ antibodies against MAP were observed to be 7%, 20% and 4.4%. No significance differences in these titers were observed among the three institutes. The IgG, IgG_1_ and IgG_2_ antibodies in serum obtained from a veterinary hospital were observed to be 55.3%, 42% and 42%, respectively. Significant differences were found between the non-clinical and clinical samples. The titers in the clinical samples showed a high degree of variance, whereas low variance was found in the non-clinical samples. The IgG antibody levels were thought to be induced following exposure to MAP-contaminated feed. The difference in titers between the clinical and non-clinical samples is likely to be related to the amount of MAP antigen contamination in dog foods.

## 1. Introduction

Johne’s disease (JD) arises from infection with *Mycobacterium avium* subsp. *paratuberculosis* (MAP) and produces chronic granulomatous gastroenteritis in cattle [1,2,3,4]. JD is a severe disease in cattle that leads to serious economic losses in the livestock industry [4,5]. JD is an infectious disease that is subject to a government policy that requires reporting its occurrence in domestic animals, and is well controlled in Japan as a result [6,7]. Moreover, the relationship between MAP and Crohn’s disease (CD), inflammatory bowel diseases (IBD), Type I diabetes and multiple sclerosis in humans has been recognized [8,9,10,11,12,13]. Furthermore, MAP has been isolated from intestinal tissue of humans with CD and IBD [14]. Moreover, specific antibodies against MAP have been detected in Japanese patients with CD [15,16,17]. IgG antibodies, especially IgG_1_ and IgG_4_, against MAP have been detected in serum of Japanese patients [18]. These serological reports spurred discussion of the role of MAP in the etiology of these diseases. It has been suggested that the detection of MAP antibodies is likely to be attributable to MAP exposure via contaminated dairy products. The prevalence of JD is very high in countries around the world, except those countries with established MAP epidemic prevention systems, such as Japan and Sweden [7,19]. Milk or meat collected from cattle suffering from JD in areas with high infection rates are likely to be contaminated with MAP [20,21]. As an example, MAP has been detected in raw milk from dairy cattle [9,22]. Through pet food produced from cows, which is impossible to use as human food, dogs and cats may be exposed to MAP through pet food.

The relationship between MAP and various diseases has not been clarified. However, if dogs are exposed to MAP, it might trigger diseases similar to those in humans. Currently, whether dogs have been exposed to the MAP antigen has not been clarified. Therefore, it is necessary to determine the degree of MAP antigen exposure in dog populations. The aim of this study is to clarify the MAP antibody seroprevalence in dogs bred in Japan.

## 2. Materials and Methods

### 2.1. Sera

The study population comprised 92 canine sera samples (non-clinical) from beagle dogs raised at three different institutes in Japan, and 57 sera (clinical) from various dog breeds treated at veterinary hospitals in Kanagawa Prefecture. The non-clinical serum samples were collected from beagle dogs raised at three different institutes that were breeders of beagle dogs for use in laboratory experiments. The dogs from which serum was collected were healthy dogs that had not yet been used in any experiments. Clinical serum samples were collected from dogs visiting for routine testing. All experiments were approved by the Institutional Review Board of Azabu University (approval No. 200206-4, 20 February 2020).

### 2.2. Measurement of Titers against MAP

The MAP titer was measured by a modified method of Otubo et al. [18]. Fifty microliters of ethanol-extracted MAP antigens were adsorbed to a 96-well plate (Thermo Fisher Scientific, Inc., MA, USA) by overnight evaporation at room temperature. Three hundred microliters per well of 20% Blocking One^TM^ (Nacalai Tesque, Inc., Kyoto, Japan) in phosphate buffered saline (PBS, pH 7.2) was added to each well and incubated at 37 °C for 1 h. After rinsing with PBS containing 0.1% Polyoxyethylene(20)Sorbitan Monolaurate (Wako Pure Chemical Industries, Ltd., Osaka, Japan), serum samples were diluted in PBS containing 50 mg/mL of *Mycobacterium phlei*, which is used to improve specificity in the diagnosis of paratuberculosis. Fifty microliters of a serum sample were added to each well and incubated at 37 °C for 1 h. After rinsing as above, horseradish peroxidase (HRP)-labeled-goat anti-dog IgG antibody, anti-dog IgG_1_-HRP or anti-dog IgG_2_ (all HRP-labeled antibodies were purchased from Bethyl Laboratories, Inc., Montgomery, Texas, USA, and diluted 1:1000 in PBS) was added to each well and incubated at 37 °C for 1 h. After rinsing again, the substrate (ABTS) was added at 100 µL/well (Roche Diagnostics GmbH, Penzberg, Germany) and absorbance at 415 nm was measured using a microplate reader (Corona Electric Co., Ltd., Ibaraki, Japan).

### 2.3. Statistical Analysis

Statistical analysis was performed using Graphpad Prism 8.0 software (GraphPad Software, Inc., San Diego, CA, USA). The area under the curve (AUC) and optimal cut-off values were determined individually by receiver operating characteristic (ROC) curve analysis, setting the specificity at 95% and with the corresponding sensitivity chosen accordingly. Comparisons of the ELISA results between different groups was performed using the Mann–Whitney nonparametric test. *p*-values less than 0.05 were considered significant.

## 3. Results

The titers of total IgG, IgG_1_ and IgG_2_ antibodies against MAP in non-clinical samples and clinical samples are shown in Figure 1; Figure 2, respectively. The percentages were calculated by dividing the number of samples above the cutoff value by the number of samples in which the antibodies against MAP were detected. Based on the determined cut-off point, 2.4% of institute A, 20% of institute B, and 9% of institute C were positive for anti-MAP IgG antibody (Figure 1A); 7% of institute A, 20% of institute B, and 0% from institute C were positive for anti-MAP IgG_1_ antibody (Figure 1B); 7% from institute A, 20% from institute B, and 4.4% from institute C were positive for anti-MAP IgG_2_ antibody (Figure 1C). Unfortunately, only five samples could be obtained from institute B. Since IgG, IgG_1_, and IgG_2_ antibodies were detected in one of the samples, it was considered that the percentage above the cut-off value was 20%. No significant differences were observed among the samples from the three institutes.

Concerning the frequencies of antibodies in the clinical samples, 55.3% were positive for anti-MAP IgG, 42.0% for anti-MAP IgG_1_ and 42.0% for anti-MAP IgG_2_ antibodies (Figure 2). Overall, the antibody titer against MAP antigens was elevated, and the difference between the non-clinical and clinical groups was statistically significant for IgG (AUC = 0.75, *p* < 0.0001) (Figure 2A), IgG_1_ (AUC = 0.70, *p* < 0.02) (Figure 2B) and IgG_2_ (AUC = 0.75, *p* < 0.0001) (Figure 2C).

## 4. Discussion

IgG antibodies against MAP were detected in both non-clinical and clinical serum samples. IgG antibodies against MAP have also been detected in human serum in Japanese patients [7]. JD is a common disease in countries without control programs, where it is more likely to spread [1,9,22,23,24]. In Japan, more than half of the pet foods on the market are imported from overseas manufacturers [25], with imports from countries such as the United States and France increasing [25]. Dogs taken to veterinary hospitals may have been fed either domestically produced or imported pet foods by their owners. MAP, an intracellular parasite, proliferates in the intestinal mucosa, is excreted in feces, and migrates hematogenously to various systemic organs as the disease progresses. Therefore, it is possible that MAP may contaminate the pet food imported from foreign countries during the manufacturing process. Furthermore, MAP is excreted into milk in cows with JD that are otherwise asymptomatic [20,21,26,27,28]. It is quite possible that contaminated dairy products for humans also contain MAP antigens. Moreover, these dogs may have also been fed foods intended for humans, such as beef or dairy products, which have been imported from overseas countries. The family dogs were also presumed to have been fed various kinds of human foods contaminated with MAP, such as dairy products or raw meat, by each owner. Conversely, it is thought that the dogs from the three institutes were provided with generally similar diets composed of commercial solid animal foods available in Japan. Solid dog foods are typically composed of soybean meal, white fish meal, yeast and soybean oil, and they are thought to have very low levels of MAP contamination [29,30]. We observed that MAP titers were significantly higher in clinical samples than in non-clinical samples. In addition, large variations in anti-MAP titers were observed in the clinical samples. This difference is presumed to be due to the difference in the exposure dose of MAP between non-clinical and clinical samples. Therefore, it was expected that a greater variation in antibody titers would be observed in the clinical samples than in the non-clinical samples. These results also suggest that, as in humans, IgG antibodies are produced following exposure to MAP-contaminated pet foods.

High titers of IgG_1_ and IgG_4_ anti-MPA antibodies have been observed in human serum samples [18]. The presence of IgG_4_ suggests a relationship with IgG_4_-related diseases [18]. Unfortunately, a secondary antibody against IgG_4_ was not commercially available at the time of this study. The titers of IgG_2_ were found to be higher than those of IgG_1_, but this result is not obviously associated with any specific diseases. Further studies will be needed to clarify the relationship between MAP and diseases in dogs.

## 5. Conclusions

Antibodies against MAP were detected in serum samples from dogs raised in Japan. These findings suggest that dogs raised in Japan are exposed to MAP through contaminated pet food. The seroprevalence and antibody titers were higher, with large variations, in samples obtained from veterinary clinics than in non-clinical samples. This difference in seroprevalence might be related to differences in the dietary history of dogs reared in institutions and in homes.

## Figures and Tables

**Figure 1 vetsci-07-00093-f001:**
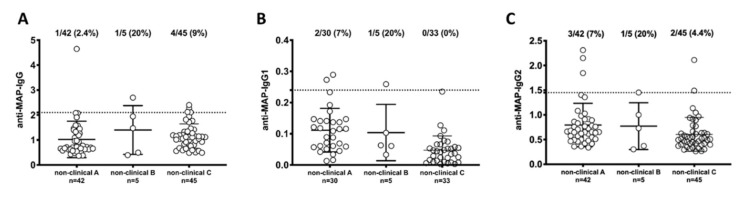
Anti-*Mycobacterium avium* subsp. *paratuberculosis* (MAP) titers in serum from dogs raised at three non-clinical institutes in Japan. (**A**) titers anti-MAP-IgG, (**B**) titers anti-MAP-IgG_1_, (**C**) titers anti-MAP-IgG_2_. Dot plot showing the distribution of total IgG detected by ELISA. The horizontal bars represent the median plus interquartile range, while the percent fraction of antibody positive sera is indicated on top of distribution. Cut-off values for positivity, calculated by ROC analysis, are indicated by dashed lines.

**Figure 2 vetsci-07-00093-f002:**
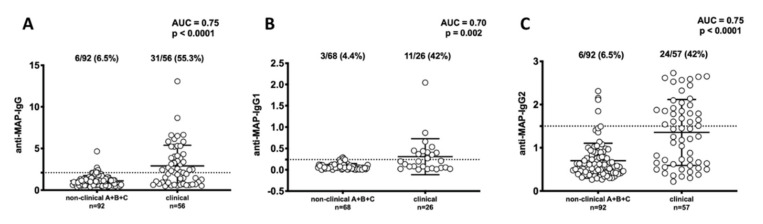
Anti-*Mycobacterium avium* subsp. *paratuberculosis* (MAP) titers in serum from dogs raised in Japan. Non-clinical samples were obtained from three institutes and clinical samples were obtained from veterinary hospitals in Japan. (**A**) titers anti-MAP-IgG, (**B**) titers anti-MAP-IgG_1_, (**C**) titers anti-MAP-IgG_2_. Dot plot showing the distribution of total IgG detected by ELISA. The horizontal bars represent the median plus interquartile range, while the percent fraction of antibody positive sera, area under ROC curve (AUC) and *p*-values are indicated on top of distribution. The dotted line lines designate the cutoff for positivity used in each assay, as calculated by ROC analysis.

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
