# Peer review of "Seroprevalence of Immunoglobulin G Antibodies Against Mycobacterium avium subsp. paratuberculosis in Dogs Bred in Japan"

_vetsci, 2020, doi:10.3390/vetsci7030093_

Round 1
Reviewer 1 Report
Dear authors,
congratulations for your report. Here are my comments/suggestions/corrections:
INTRODUCTION
(1) Lines 40 and 41: should write "of human with CD and IBD".
(2) Line 43: erase the word "sero-reports" and replace for "serological reports".
MATERIAL AND METHODS
(1) Line 62: write "The measurement of MAP"
RESULTS
(1) Figure 1. You should name each graphic as A, B, and C respectively. There is an error in figure 1C: you should replace "anti-MAP-IgG1" for "anti-MAP-IgG2".
(2) Figure 2. You should name each graphic as A, B, and C respectively.
DISCUSSION
(1) You should state if you considered the Beagle dog populations as healthy dogs or not. If that is so, what measures did you take to state that the non-clinical dogs were healthy?
(2) Clinical dogs. You should say something about their health status. Were dogs that were admitted to the clinics for routine checks, maybe sterilization procedures, or serious diseases?
(3) Do you think that the process of industrialized pet food making does not destroy MAP antigens? If that so, can you add proofs/references on this matter?
(4) Non-clinical dogs. You should be aware of the food record of these dogs. If they are experimental dogs, a food record should be present. You don't mention any explanation on the different results between clinical and non-clinical dogs; so, taking into account the results, the reader could infer that clinical dogs were "more positive" just because they also eat home-prepared food.
Thank you, that's all for the moment.
Author Response
Reply for Reviewer 1
Introduction
- Revised as reviewer pointed out.
- Revised as reviewer pointed out.
Material and Methods
- Revised as reviewer pointed out.
Figure 1.
- Revised as reviewer pointed out.
Figure 2
- Revised as reviewer pointed out.
Discussion
- The sera tested as non-clinical in this study were obtained from beagle dogs raised as experimental animals in three institutes. In addition, these beagle dogs have not been used in experiments before their serum is collected. These descriptions have been added to the revised manuscript.
- Serum tested as a clinical sample was collected for routine checking at a veterinary hospital. The remaining samples after routine checks were dispensed and used for this study.
- MAP is an intracellular parasite and is distributed throughout the bovine body. MAP has also been detected in pasteurized milk in countries with high Johne's disease (JD) contamination. Therefore, it is considered that MAP is contained as an antigen in particularly foreign pet food made from cows suffering from these JD.
The positive rate of antibodies to MAP in clinical samples was higher than in non-clinical samples. It is considered that the family dogs eats various types of pet food, including foreign pet food. In addition, family dogs are very likely to be fed human food that is thought to contain MAP. On the other hand, beagle dogs collected non-clinical samples were fed a solid diet made from soybean meal, whitefish meal, yeast, and soybean oil, and are considered to have very low levels of MAP contamination. It is considered that the difference in the amount of exposure to MAP due to this difference in feed is due to the difference in the positive rate for MAP.
Reply for Reviewer 1
Introduction
- Revised as reviewer pointed out.
- Revised as reviewer pointed out.
Material and Methods
- Revised as reviewer pointed out.
Figure 1.
- Revised as reviewer pointed out.
Figure 2
- Revised as reviewer pointed out.
Discussion
- The sera tested as non-clinical in this study were obtained from beagle dogs raised as experimental animals in three institutes. In addition, these beagle dogs have not been used in experiments before their serum is collected. These descriptions have been added to the revised manuscript.
- Serum tested as a clinical sample was collected for routine checking at a veterinary hospital. The remaining samples after routine checks were dispensed and used for this study.
- MAP is an intracellular parasite and is distributed throughout the bovine body. MAP has also been detected in pasteurized milk in countries with high Johne's disease (JD) contamination. Therefore, it is considered that MAP is contained as an antigen in particularly foreign pet food made from cows suffering from these JD.
The positive rate of antibodies to MAP in clinical samples was higher than in non-clinical samples. It is considered that the family dogs eats various types of pet food, including foreign pet food. In addition, family dogs are very likely to be fed human food that is thought to contain MAP. On the other hand, beagle dogs collected non-clinical samples were fed a solid diet made from soybean meal, whitefish meal, yeast, and soybean oil, and are considered to have very low levels of MAP contamination. It is considered that the difference in the amount of exposure to MAP due to this difference in feed is due to the difference in the positive rate for MAP.
Reply for Reviewer 1
Introduction
- Revised as reviewer pointed out.
- Revised as reviewer pointed out.
Material and Methods
- Revised as reviewer pointed out.
Figure 1.
- Revised as reviewer pointed out.
Figure 2
- Revised as reviewer pointed out.
Discussion
- The sera tested as non-clinical in this study were obtained from beagle dogs raised as experimental animals in three institutes. In addition, these beagle dogs have not been used in experiments before their serum is collected. These descriptions have been added to the revised manuscript.
- Serum tested as a clinical sample was collected for routine checking at a veterinary hospital. The remaining samples after routine checks were dispensed and used for this study.
- MAP is an intracellular parasite and is distributed throughout the bovine body. MAP has also been detected in pasteurized milk in countries with high Johne's disease (JD) contamination. Therefore, it is considered that MAP is contained as an antigen in particularly foreign pet food made from cows suffering from these JD.
The positive rate of antibodies to MAP in clinical samples was higher than in non-clinical samples. It is considered that the family dogs eats various types of pet food, including foreign pet food. In addition, family dogs are very likely to be fed human food that is thought to contain MAP. On the other hand, beagle dogs collected non-clinical samples were fed a solid diet made from soybean meal, whitefish meal, yeast, and soybean oil, and are considered to have very low levels of MAP contamination. It is considered that the difference in the amount of exposure to MAP due to this difference in feed is due to the difference in the positive rate for MAP.
Reply for Reviewer 1
Introduction
- Revised as reviewer pointed out.
- Revised as reviewer pointed out.
Material and Methods
- Revised as reviewer pointed out.
Figure 1.
- Revised as reviewer pointed out.
Figure 2
- Revised as reviewer pointed out.
Discussion
- The sera tested as non-clinical in this study were obtained from beagle dogs raised as experimental animals in three institutes. In addition, these beagle dogs have not been used in experiments before their serum is collected. These descriptions have been added to the revised manuscript.
- Serum tested as a clinical sample was collected for routine checking at a veterinary hospital. The remaining samples after routine checks were dispensed and used for this study.
- MAP is an intracellular parasite and is distributed throughout the bovine body. MAP has also been detected in pasteurized milk in countries with high Johne's disease (JD) contamination. Therefore, it is considered that MAP is contained as an antigen in particularly foreign pet food made from cows suffering from these JD.
The positive rate of antibodies to MAP in clinical samples was higher than in non-clinical samples. It is considered that the family dogs eats various types of pet food, including foreign pet food. In addition, family dogs are very likely to be fed human food that is thought to contain MAP. On the other hand, beagle dogs collected non-clinical samples were fed a solid diet made from soybean meal, whitefish meal, yeast, and soybean oil, and are considered to have very low levels of MAP contamination. It is considered that the difference in the amount of exposure to MAP due to this difference in feed is due to the difference in the positive rate for MAP.

Reviewer 2 Report
In the study „Seroprevalence of immunoglobulin G antibodies against Mycobacterium avium subsp: paratuberculosis in dogs bred in Japan“, Kuribayashi et al. analyzed non-clinical and clinical samples from dogs bred in Japan for their presence of immunoglobulin G antibodies against Mycobacterium avium subsp: paratuberculosis (MAP) in ELISA. Authors found significant differences between non-clinical and clinical samples with IgG, IgG1 and IgG2 antibodies in serum from clinical samples being elevated. Reason for this is speculated to be exposure to contaminated feed.
In general, the study adds interesting data towards the seroprevalence status of dogs in Japan and thus, the overall situation that might arise due to import of contaminated food. However, I think the manuscript needs some major adjustments before suitable for publication:
Major points:
Please clarify in the manuscript and correct (inconsistent throughout the manuscript): 92 non-clinical samples from 3 institutes (as stated in the abstract) or 71 non-clinical samples as stated in Materials and Methods. The same is true for the clinical samples: In the Abstract it says 57 samples from a (1?) veterinary hospital versus 87 clinical samples treated at veterinary hospitals (several) in Methods or in l. 104.
For all non-clinical samples, 20% of IgG1 antibodies could be determined (compared to low or no IgG and IgG2 antibodies). In contrast, for the clinical samples higher total IgG was determined and equal % of IgG1 and IgG2. Can you comment/speculate on the difference of IgG’s produced?
The conclusion that difference in titers between clinical and non-clinical samples is likely to be related to the amount of MAP antigen contamination in dog food? Can one draw the conclusion that healthy dogs (samples from insitutes) get non-contaminated food whereas dogs that were sick (veterinary hospital) were fed with contaminated food? Also, is there anything know about the conditions why dogs where brought to the hospital (all food-related diseases or also other diseases)?
In the Introduction (ll. 49-51), the two sentences are a contradiction or at least, the logic rational is difficult to follow. Dog/cat food comes from infected livestock, that is not available for human food. But both humans and cats/dogs are exposed through food…?
In l. 52: Authors state that „few“ reports have evaluated exposure of dogs to MAP antigen. Please add references.
With regards to the aim: Why is the seroprevalence of MAP in Japanese dogs or relevance? Authors should add a sentence to explain the relevance for the reader (zoonotic risk?).
Authors compare non-clinical beagle dog samples with various breed samples from hospitals. Were beagles under the different breeds as well? Are beagles more prone to develop IgG1 antibodies than other breeds (looking at the data sets for the different institutes and IgG’s, where always IgG1 comes up)? Are beagles equally susceptible to MAP? Are they fed the same food or do dogs at the institutes receive special treated/examined food?
For testing of the samples, authors refer to previous publication. Most differences are indicated in the manuscript. However, for the reader, it would be important to know if samples are tested undiluted or diluted (l. 68); what is the company of the secondary antibodies and how are they diluted. What controls are used in the assay (positive control?)? And how is the background/cut-off determined?
Fig 1: Why does number of samples from the respective institutes vary depending on determination of IgG, IgG1 or IgG2 antibodies? Samples from B remain stable (n=5), but samples from A are n=42 for IgG, n=30 for IgG1 and n=42 for IgG2 and the same goes for samples from C (n=45, n=33, n=33). Is there a reason for that? Please address. Looking at the graphs, is there really a positive IgG2 samples for B (especially if compared ot IgG1 from C, where the dot on the cut-off line is not counted)? Indicating the positives for each institutes and each IgG/ subclass is fine, but adding the % for e.g. for 1 out of 5 (which is 20%) gives the impression of a high prevalence (but the sample number is quite low). It would be better to leave these numbers out and just refer to the total number for IgG, IgG1 and IgG2 (e.g. 6 out of 92 samples are positive for IgG > results in 6.5% positives) as depicted in Fig. 2. Are the respective positive samples for IgG, IgG1 and IgG2 the same throughout the testing?
Fig 2: Please explain n=92 for IgG2? In Fig. 1 only n=80 samples are tested. Also, why is there a variation between n=56, n=26 and n=57 for the clinical samples. Is there a selection criteria?
Please revise your discussion and structure your statements. The discussion goes from MAP antigens in humans to contaminated food, to food possibly fed to dogs, imported food, to prevalence of JD, to control of JD, back to contaminated food and food possibly fed to animals. Please streamline, order and avoid repetitions.
Why was then IgG2 determined if IgG4 would have been the interesting one?
The conclusion drawn from the data (amount of MAP antigen related to contamination of dog food) should be explained more clearly.
Minor points:
Please correct spelling mistakes: l. 64 > delete hyphen (three hundred); l.102 > delete extra space after anti- ; l. 134 > MAP, l.135 > related; l. 137 > associated
Are samples tested in duplicate/triplicate?
Fig 1: Labelling of the y-axis in the third graph should read IgG2 and not IgG1
In l. 91, l. 102: is „raised“ the appropriate word? Dogs are raised, serum i staken.
in l. 110: Population?
in l. 136 > a secondary anti-dog?
Author Response
Reply for reviewer 2
Major points
- The numbers of non-clinical samples were 92 and clinical samples were 57. We revised correct number in Abstract, Materials and Methods.
- The percentage of IgG1 antibody is 20%, which is higher in institute B than in the other two institutes. Unfortunately, only five samples were collected from institute B. Only one sample exceeded the cut-off value, indicating a higher value than other institutes, which was not considered to be a particularly high proportion.
The percentage of IgG1 and IgG2 antibodies were equal in clinical sample. The titers of IgG2 antibody against MAP were higher than IgG1 IgG1 antibody was detected in samples with high IgG2 antibody titers. It is presumed that IgG2 antibodies account for the majority of IgG antibodies. - The relationship between MAP and diseases such as CD has not been clarified. This relationship is under debate. MAP is speculated to trigger the onset of the disease. IgG4 antibody was detected in Japanese. Otsubo et al. presumed a relationship between MAP and IgG4-related diseases in humans. In this study, IgG antibodies against MAP were detected in dog serum. The results suggested that dogs are exposed through MAP-contaminated pet food. However, further research is needed to clarify the relationship between MAP and human-like canine illness.
- Cows infected with Johne's disease (JD) cannot be used for human consumption. However, MAP was detected in humans such as CD. In addition, MAP has been reported to be detected in milk despite being sterilized. In addition, since antibodies against MAP were detected in Japanese serum, it was considered that humans were exposed via foods contaminated with MAP.
On the other hand, these cows infected with JD can be used as ingredients for pet food. Dogs and cats may then be exposed via pet food and serum titers of antibodies to MAP. - The expression reviewer pointed out, Few reports..., was inappropriate. The prevalence of antibodies against MAP has not been evaluated yet. Then, this description was revised.
- Prevalence of antibodies to MAP in dogs has not been evaluated. Dogs may have been fed pet food that may be contaminated with MAP or human food contaminated with MAP by their owners. Therefore, dogs may be exposed to MAP through food similar to humans. Moreover, when antibodies against MAP are detected in the serum of dogs, it is possible that MAP triggers the disease as in humans. Then, we evaluated the serum titer of antibodies to dog MAP.
- Beagle dogs raised institutes were fed a solid feed marketed in Japan. The family dogs weren't given a special diet, such as pet food manufactured in foreign countries. The antibody titer against MAP in institute B was higher than that in institutes A and C. Unfortunately, only five samples could not be purchased from institutes B. The IgG, IgG1, and IgG2 antibody titers in one of these samples were high. Therefore, the percentage above the cutoff value was 20%. We do not consider that IgG1 antibodies were more easily produced than IgG2 antibodies from this percentage.
- The antibody titer to MAP was measured by the modified method of Otubo et al. This document is cited. All secondary antibodies, HRP-labeled antibodies, were purchased from Bethyl Laboratories, Inc. These secondary antibodies were diluted in PBS.
- The parameters shown in the figure indicate the number of samples above the cutoff. It's certainly misleading, so I've added a description to Legend.
- The parameters shown in the figure indicate the number of samples above the cutoff. It's certainly misleading, so I've also added a description to Legend.
- The discussion has been completely revised. The antibodies against MAP were detected from serum of dogs raised in Japan. We considered that dogs, especially family dogs, were likely to be fed with pet foods from abroad that were not control JD in farm or human foods contaminated with MAP.
- Previously reported human evaluations have examined all four subclasses of IgG. However, unfortunately, no antibodies against IgG1 and IgG2 are commercially available as secondary antibodies in dogs. Therefore, we evaluated these two subclasses.
- The conclusion has been completely revised.
Minor points
- Q) Are samples tested in duplicate/triplicate.
A) The absorbance was measured with a single in each sample. Because, Clinical samples were rest sera used for routine testing and donated by the veterinary hospital. Therefore, some specimens were very small, and could not be measured by duplicate. Then, after confirming the reproducibility of the absorbance of the same sample, the measurement was performed with a single.
Other minor points were revised as reviewer pointed out.

Round 2
Reviewer 2 Report
Authors have revised their manuscript and addressed most points in the revised manuscritpt and the rebuttal. However, certains mistakes that have been highlighted in the previous review are still present in the manuscript and should be corrected (e.g. sample numbers and some minor points).
Author Response
Dear Reviewer 2
The description has been revised so that the number of samples used in this study (Line 58).
The authors revised the legend regarding clarification of the number of samples shown in the figures. The revised parts are marked with yellow markings.
